# Phenol-Soluble Modulin α3 Stimulates Autophagy in HaCaT Keratinocytes

**DOI:** 10.3390/biomedicines11113018

**Published:** 2023-11-10

**Authors:** Áron Dernovics, György Seprényi, Zsolt Rázga, Ferhan Ayaydin, Zoltán Veréb, Klára Megyeri

**Affiliations:** 1Department of Medical Microbiology, Albert Szent-Györgyi Medical School, University of Szeged, Dóm tér 10., H-6720 Szeged, Hungary; dernovics.aron@med.u-szeged.hu; 2Department of Anatomy, Histology and Embryology, Albert Szent-Györgyi Medical School, University of Szeged, Kossuth L. sgt. 40., H-6724 Szeged, Hungary; seprenyi.gyorgy@med.u-szeged.hu; 3Department of Pathology, University of Szeged, Állomás u. 2, H-6720 Szeged, Hungary; razga.zsolt@med.u-szeged.hu; 4Hungarian Centre of Excellence for Molecular Medicine (HCEMM) Nonprofit Ltd., Római krt. 21., H-6723 Szeged, Hungary; ferhan.ayaydin@hcemm.eu; 5Laboratory of Cellular Imaging, Biological Research Centre, Eötvös Loránd Research Network, Temesvári krt. 62., H-6726 Szeged, Hungary; 6Regenerative Medicine and Cellular Pharmacology Laboratory, Department of Dermatology and Allergology, University of Szeged, Korányi Fasor 6, H-6720 Szeged, Hungary; vereb.zoltan@med.u-szeged.hu; 7Biobank, University of Szeged, H-6720 Szeged, Hungary; 8Interdisciplinary Research Development and Innovation Center of Excellence, University of Szeged, H-6720 Szeged, Hungary

**Keywords:** pore-forming toxin, staphylococci, phenol-soluble modulin, autophagy, keratinocyte

## Abstract

Background: Phenol-soluble modulins (PSMs) are pore-forming toxins (PFTs) produced by staphylococci. PSMs exert diverse cellular effects, including lytic, pro-apoptotic, pro-inflammatory and antimicrobial actions. Since the effects of PSMs on autophagy have not yet been reported, we evaluated the autophagic activity in HaCaT keratinocytes treated with recombinant PSMα3. Methods: The autophagic flux and levels of autophagic marker proteins were determined using Western blot analysis. Subcellular localization of LC3B and Beclin-1 was investigated using an indirect immunofluorescence assay. The ultrastructural features of control and PSMα3-treated cells were evaluated via transmission electron microscopy. Cytoplasmic acidification was measured via acridine orange staining. Phosphorylation levels of protein kinases, implicated in autophagy regulation, were studied using a phospho-kinase array and Western blot analysis. Results: PSMα3 facilitated the intracellular redistribution of LC3B, increased the average number of autophagosomes per cell, promoted the development of acidic vesicular organelles, elevated the levels of LC3B-II, stimulated autophagic flux and triggered a significant decrease in the net autophagic turnover rate. PSMα3 induced the accumulation of autophagosomes/autolysosomes, amphisomes and multilamellar bodies at the 0.5, 6 and 24 h time points, respectively. The phospho-Akt1/2/3 (T308 and S473), and phospho-mTOR (S2448) levels were decreased, whereas the phospho-Erk1/2 (T202/Y204 and T185/Y187) level was increased in PSMα3-treated cells. Conclusions: In HaCaT keratinocytes, PSMα3 stimulates autophagy. The increased autophagic activity elicited by sub-lytic PSM concentrations might be an integral part of the cellular defense mechanisms protecting skin homeostasis.

## 1. Introduction

Staphylococci are important human pathogenic bacteria. The coagulase-positive, highly virulent *Staphylococcus aureus* (*S. aureus*) is a well-known cause of various infections and toxin-mediated diseases, including skin and soft tissue infections, pneumonia, osteoarticular infections, endocarditis, meningitis, sepsis, hospital infections, toxic shock syndrome, food poisoning and staphylococcal scalded skin syndrome [1,2,3]. *S. aureus* possesses exceptional adaptive potential, mainly due to its multifactorial virulence. The immunomodulatory structural components, the broad array of secreted enzymatic virulence factors and various exotoxins promote the establishment of severe, devastating and frequently lethal *S. aureus* infections [1,2,3]. The coagulase-negative staphylococci (CoNS), such as members of the *S. epidermidis* group and *S. lugdunensis*, colonize the human skin and mucous membranes. CoNS have a more limited range of virulence factors than that of *S. aureus*. *S. epidermidis*, as a member of the normal cutaneous microbiota, contributes to maintaining skin colonization resistance and barrier integrity [4,5]. However, invasive CoNS infections can occur in cases of impaired barrier functions and other predisposing factors [4,5]. CoNS can cause various systemic infections related to the implantation of medical devices, such as prosthetic vascular grafts, prosthetic heart valves, pacemakers, coronary stents, ventriculoperitoneal shunts and cerebrospinal fluid shunts [4,5].

Phenol-soluble modulins (PSMs) are pore-forming toxins (PFTs) produced by both coagulase-positive staphylococci (CoPS) and CoNS [6,7]. Members of the *S. aureus*-related complex (*S. aureus*, *S. argenteus* and *S. schweitzeri*) and the coagulase-negative *S. epidermidis*, *S. lugdunensis*, *S. capitis* and *S. saccharolyticus* were shown to produce PSMs [8,9]. Almost all known PSM peptides are encoded by genes located in the bacterial genome or a mobile genetic element present in virtually all staphylococcal species [10]. The extent and pattern of PSM production, however, show different characteristics specific to the bacterial species, strain and circumstances of the infection [10]. Production of PSMs is regulated by the accessory gene regulator (Agr), staphylococcal accessory regulator (SarA) and LuxS/autoinducer-2 systems [11]. In the early stages of infection, when bacterial numbers are low, the Agr quorum sensing system is inactive, and therefore, the production of many enzymatic virulence factors and toxins, including PSMs, is reduced [11]. In the late stages of infection, the production of these virulence factors rises sharply, parallel to the increase in bacterial cell densities [11]. PSMs exert diverse biological effects. Most PSMs, particularly the α-types, have cytolytic activity against various cell types, including erythrocytes, granulocytes, monocytes, endothelial and epithelial cells and osteoblasts [12,13]. PSMs, in μM concentrations, assemble into pores on the cytoplasmic membrane and in organelles belonging to the endomembrane system. The toxin-mediated disintegration of the cytoplasmic membrane leads to cell demise, whereas intracellularly acting PSMs promote the survival of bacteria within cells by damaging the endosomal/phagosomal membrane [7,10,14]. Moreover, PSMs, produced by *S. epidermidis*, can also disrupt bacterial membranes, thereby exerting a selective antimicrobial action against the highly pathogenic *S. aureus* and *Streptococcus pyogenes* [15,16]. These data strongly support that, when present on intact skin, *S. epidermidis* PSMs can confer some protection against skin infections caused by these highly invasive bacteria [15]. Finally, PSMs exert a powerful immunomodulatory effect triggered by their binding to formyl peptide receptor 2 (FPR2) [17]. FPR2 is a G protein-coupled 7-transmembrane receptor that senses PSMs in nM concentrations [17]. Toll-like receptor 2 (TLR2), acting in concert with TLR6, was also shown to be capable of transducing cellular responses to PSMs [18]. The effect of PSMs on the immune response is multifaceted. These exotoxins stimulate the production of various cytokines, chemokines and cell adhesion molecules. In keratinocytes, PSMα3 upregulates the synthesis of CXCL1, CXCL2, CXCL3, CXCL5, CXCL8, CCL20, interleukin-1α (IL-1α), IL-1β, IL-6, IL-36γ and tumor necrosis factor-α (TNF-α) [19]. Moreover, PSMs exert a chemoattractant effect on neutrophil granulocytes and stimulate the formation of neutrophil extracellular traps [20]. However, PSMs are also key elements of the immune escape strategies employed by staphylococci. PSMs help staphylococci evade the immune system by destroying neutrophil granulocytes [7,10]. They also impair the function of dendritic cells and thereby shift T-cell differentiation toward the formation of Treg cells, which are rather ineffective in eliminating staphylococci from the body [21]. An interesting study demonstrated that *S. aureus* strains with high Agr activity levels block autophagic flux and lead to the accumulation of autophagosomes, which in turn provide a protected environment for bacteria in murine-bone-marrow-derived macrophages [22]. All these mechanisms promote the survival of these bacteria within the body. 

During infection, many cells are exposed to sub-lytic doses of staphylococcal PFTs, which allow them to activate cell recovery pathways. PFT pores can be repaired or eliminated by various mechanisms. Annexins can repair plasma membrane damage by clogging pores [23]. The shedding of microvesicles containing toxin pores or incomplete pore-forming structures can remove the pores [23]. Plasma membrane homeostasis can also be restored by the internalization of PFT pores. The endocytosed PFT pores are then transported to multivesicular bodies (MVBs) and degrade after fusion with lysosomes [23]. Besides membrane repair mechanisms, several other recovery pathways are activated, including the mitogen-activated protein kinase (MAPK) survival pathway, cytoskeleton remodeling and autophagy [23]. 

Autophagy is a conserved catabolic process that can be divided into five phases: (i) initiation, (ii) phagophore nucleation, (iii) elongation and autophagosome formation, (iv) autophagosome-lysosome fusion and (v) cargo degradation [24]. The mechanistic target of rapamycin complex 1 (mTORC1) is a key player in the regulation of autophagy [25,26]. The inhibition of mTORC1 promotes autophagy by activating the class III phosphatidylinositol 3-kinase (PI3KC3) complex, composed of VPS34, VPS15, ATG14L and Beclin-1 [26,27]. PI3KC3 generates phosphatidylinositol 3-phosphate (PI3P) at specific membrane sources, which results in the formation of a phagophore [27,28,29]. A covalent bond is formed between the membrane-localized phosphatidylethanolamine (PE) and the microtubule-associated protein 1 light chain 3B-I (LC3B-I) protein by two ubiquitin-like conjugation mechanisms during phagophore elongation [28,29]. The PE-conjugated LC3B-I, also known as LC3B-II, is necessary for the phagophore’s elongation and autophagosome formation [30,31,32]. The autophagosomes are then transported to lysosomes and undergo fusion, resulting in autolysosomes. The cargo is subjected to degradation by lysosomal enzymes, completing the autophagic process [24]. Autophagy plays an important role in maintaining the homeostasis of human tissues and organs. This process is involved in keratinocyte differentiation, wound healing, local immune protection and effective skin barrier formation [33,34,35,36,37]. During colonization and infection, staphylococci efficiently modulate autophagy [38,39,40]. The intracellular infection of many cell types, including epithelial cells, endothelial cells and leukocytes, activates autophagy, killing some bacteria [41]. Although autophagy contributes to the elimination of staphylococci, highly virulent *S. aureus* strains producing high levels of Agr-dependent toxins are destroyed by pyroptosis more efficiently [42,43]. The caspase-1-mediated release of interleukin-1 (IL-1) attracts neutrophil granulocytes, stimulates the killing activity of phagocytes and facilitates the elimination of bacteria [41]. In contrast, the less-toxigenic *Staphylococcus* strains have low cytotoxicity and pyroptosis-inducing ability. Therefore, in infections caused by these less virulent strains, autophagy promotes the development of an intracellular milieu that is beneficial for the bacterium and persistent infection [41,42]. Previous interesting studies have already demonstrated the pro-autophagic effects of some complex TLR-2 agonist structural components of staphylococci, such as peptidoglycan and lipoteichoic acid [44,45,46]. Moreover, other interesting studies have revealed that *S. aureus* α-toxin causes potassium and ATP efflux, which, in turn, triggers a starvation response, metabolic reprogramming and increased autophagic activity elicited by mTORC1 deactivation [47,48,49]. Although it is well known that bacterial PFTs activate the autophagic cascade, investigations of the ability of staphylococcal PSMs to modify autophagy have not yet been reported. As strains of PSM-producing staphylococci colonizing or infecting the skin play a critical role in the physiological and pathological processes of the skin, we studied how PSMα3 affects autophagy in the HaCaT keratinocyte cell line. 

## 2. Materials and Methods

### 2.1. Chemical Compounds

PSMα3 (IBT Bioservices, Rockville, MD, USA) was prepared in sterile distilled water and used at 0.25, 0.5 and 1 μg/mL concentrations. The half-maximal cytotoxic concentration of the PSM-α3 lot (2306001) used was 127 µg/mL (IBT Bioservices, Rockville, MD, USA).

The autophagy inhibitor bafilomycin A1 (BFLA) (Santa Cruz Biotechnology, Dallas, TX, USA) was dissolved in dimethylsulfoxide and used at a concentration of 100 nM in all experiments. 

### 2.2. Cell Culture

The HaCaT keratinocyte cell line, kindly provided by Dr. N.E. Fusenig (Heidelberg, Germany), was cultured in Dulbecco’s modified Eagle’s medium F-12 with stable glutamine (Avantor, Radnor, PA, USA) supplemented with 10% fetal calf serum (Lonza, Verviers, Belgium) and 1% antibiotic/antimycotic (AB/AM) solution (Lonza) at 37 °C in a 5% CO_2_ atmosphere. The fetal calf serum was heat inactivated for 30 min (min) at 56 °C.

### 2.3. Immunofluorescence Assay

The control and PSMα3-treated cells were collected via trypsinization, washed and centrifuged to the surface of SuperFrost Plus glass slides (VWR, Radnor, PA, USA) at 1000 rpm for 3 min by using a Cellspin III CYTO cytocentrifuge (Tharmac GmbH, Limburg an der Lahn, Germany). Cytospin cell preparations were fixed in a mixture of methanol and acetone (1:1) and incubated at −20 °C for 10 min. To block the non-specific binding of antibodies, the cells were treated with 1% bovine serum albumin in phosphate-buffered saline (PBS) for 30 min at 37 °C. For LC3B detection, the slides were stained with a rabbit polyclonal antibody to LC3B (Merck KGaA, Darmstadt, Germany) at a 1:150 dilution and incubated for 1 h at 37 °C. For Beclin-1 detection, the slides were stained with a rabbit polyclonal antibody to Beclin-1 (Merck KGaA) at a 1:100 dilution and incubated for 1 h at 37 °C. After washing with PBS, the samples were incubated with a CF488A-conjugated anti-rabbit antibody (Merck KGaA) at a 1:300 dilution for 1 h at 37 °C. Confocal microscopy was performed using an Olympus FV1000 confocal laser scanning microscope with a UPLSAPO 60X (N.A. 1.35) oil immersion objective and 488 nm laser excitation with a 500–600 nm detection range. Background level subtraction, intensity threshold adjustment and automatic quantification of LC3B-positive vacuoles were performed using ImageJ software 1.54d (USA National Institutes of Health, Bethesda, MD, USA) [50]. The numbers of cells in each field were used to normalize the numbers of LC3B-positive puncta [51]. For every condition, 500 cells on average were examined. The fluorescence intensity of LC3B was measured, and three-dimensional surface plots were generated to depict the intensity values across the entire image by using the surface plot functions of ImageJ software [50]. To measure the fluorescence signal intensities of cells, the mean fluorescence intensity method (MFI) was used. To outline the cells, ImageJ software was used, and the MFI was determined [50]. To calculate the corrected total cell fluorescence (CTCF), the following equation was used: integrated density—(area of the selected cell × mean fluorescence of background readings).

### 2.4. Detection of Acidic Vesicular Organelles

The cells, grown on glass coverslips, were incubated for 20 min at 37 °C in a serum-free medium containing 2 μg/mL 3,6-bis(dimethylamine)acridine. After the cells had been washed, fluorescent micrographs were obtained using a Nikon ECLIPSE Ts2 Inverted LED Phase Contrast Microscope equipped with a 385/470/590 nm filter set. To detect green fluorescence, an excitation filter of 470 nm with a 534/550 nm emission filter was used. To detect red fluorescence, an excitation filter of 590 nm with a 652/665 nm emission/barrier filter was used. (Nikon, Tokyo, Japan). Acidic vesicular organelles (AVOs) were automatically quantified in each field by subtracting the background level and establishing an intensity threshold using ImageJ software [50]. 

### 2.5. Western Blot

The cells were homogenized using the CytoBuster lysis buffer (Merck KGaA). The resulting mixture was centrifuged at 10,000× *g* for 10 min to remove cellular debris. Protein concentrations in the cell lysates were determined using the Bio-Rad protein assay (Bio-Rad Laboratories Inc., Hercules, CA, USA). Supernatants were combined with a Laemmli sample buffer and boiled for 5 min. Aliquots of the supernatants were separated with 12% SDS-PAGE by using a PROTEAN II xi vertical electrophoresis chamber (Bio-Rad, Hercules, CA, USA) and transferred onto Immun-Blot polyvinylidene difluoride (PVDF) membranes (Bio-Rad) by using a Trans-Blot transfer cell (Bio-Rad). The membranes were blocked for 1 h at room temperature with a solution of PBS containing 0.05% Tween 20 and 5% dried non-fat milk (Difco Laboratories Inc., Detroit, MI, USA). Pre-blocked blots were incubated with specific antibodies for 4 h in PBS containing 0.05% Tween 20, 1% dried non-fat milk and 1% bovine serum albumin (Merck KGaA). The dilutions of rabbit anti-LC3B (Merck KGaA, L7543), rabbit anti-β-actin (Merck KGaA, A2066), rabbit anti-mTOR (GeneTex, Hsinchu, Taiwan, ROC, GTX101557) and mouse anti-phospho-mTOR (Ser2448) (Proteintech Germany GmbH, Planegg-Martinsried, Germany, 67778-1-Ig) primary antibodies were 1:1000. The blots were then incubated for 1.5 h with peroxidase-conjugated anti-rabbit and anti-mouse antibodies (Merck KGaA, A0545 and A0168, respectively). The detection of membranes was performed with a chemiluminescence detection system (Merck KGaA, Cytiva RPN2236). The autoradiographs were scanned with a GS-800 densitometer (Bio-Rad). The blots were scanned, and the relative band intensities were quantified using ImageJ software [50]. To quantify experimental LC3B-II, mTOR and phosphor-mTOR signal intensity values, Western blot normalization was performed. First, a normalization factor was determined for each lane. To determine the lane normalization factor, the value of the observed signal for actin in each lane was divided by the highest observed actin signal on the blot. Normalization factors were determined for each blot. To calculate the normalized signal of each experimental target band, the observed signal intensity of each experimental target band was divided by the lane normalization factor. To investigate the autophagosome turnover, the ratios of the normalized LC3B-II levels measured in PSMα3-treated cells to those of the control were calculated and expressed as fold changes. These data were used to calculate the degree of autophagosome formation, degradation and net turnover ratios. Formation ratios were calculated by subtracting the value of LC3B-II measured in the untreated control culture from the values measured in cells incubated in the presence of BFLA. Degradation ratios represent the differences between the values of LC3B-II in cells incubated with and without BFLA. Net autophagic turnover rates in each treatment group were obtained by dividing the degradation and formation rates.

### 2.6. Phospho-Kinase Array Analysis

A human phospho-kinase array (R&D Systems Inc., Minneapolis, MN, USA, ARY003C) was used to assess the relative phosphorylation levels of 37 signaling molecules. Control cells and cultures treated with PSMα3 for 30 min and 6 h were homogenized in a lysis buffer, followed by centrifugation at 14,000× *g* for 5 min. The protein concentrations in the resulting supernatants were determined using the Bio-Rad protein assay (Bio-Rad Laboratories Inc.). Three individual samples were pooled for each treatment group. Subsequently, 300 µg of protein from each sample was incubated overnight at 4 °C with an array containing two technical replicates of each signaling molecule. After washing, the arrays were exposed to a cocktail of phospho-site-specific biotinylated antibodies for 2 h at room temperature, followed by thorough washing and incubation with streptavidin-peroxidase for 30 min at room temperature. The antibody cocktails and the streptavidin-peroxidase were used at 1:50 and 1:2000 dilutions, respectively. Signal development was achieved by using a chemiluminescence detection system. The arrays were scanned, and the spot densities of phospho-proteins were quantified using ImageJ software [50]. After subtracting background values, the quantified values were normalized to the positive controls on the same membrane. 

### 2.7. Transmission Electron Microscopy

Samples were fixed in a 0.1 M PBS-buffered 2.5% glutaraldehyde solution (pH 7.4) with 2.25% dextran for 2 h. The samples were post-fixed in a 1% OsO_4_ solution for 1 h. After dehydration in an ethanol gradient (70%, 96% and 100% ethanol for 20 min each), samples were embedded in Embed 812 (Electron Microscopy Sciences, Hatfield, PA, USA, 14120). Ultrathin sections (70 nm) were stained with 1% uranyl acetate (Electron Microscopy Sciences, 22400) and 3% lead citrate (Electron Microscopy Sciences, 22410) for 3 min sequentially. Sections were examined in a JEOL JEM-1400Plus transmission electron microscope (JEOL USA Inc., Peabody, MA, USA) at 120 kV. The cytoplasmic area was determined via the point counting method. Autophagosomes and autolysosomes were scored according to their morphology [52] and were counted. The densities of autophagosomes/autolysosomes, amphisomes and multilamellar bodies per unit of cytoplasmic area were calculated by dividing the number of vacuoles by the cytoplasmic area in the same micrograph. The results are presented as the number of organelles/cytoplasmic area ± standard deviation (SD). 

### 2.8. Statistical Analysis 

Statistical significance was analyzed with a one-way ANOVA followed by Sidak’s multiple comparisons post hoc test or an unpaired *t*-test based on similar variances. All statistical analyses were performed using GraphPad Prism 6 software (GraphPad Software Inc., San Diego, CA, USA), and *p* values less than 0.05 were considered statistically significant.

## 3. Results

### 3.1. The Effect of PSMα3 on the Autophagic Flux

To elucidate how staphylococcal PSMα3 affects basal autophagy, the levels of LC3B-I and LC3B-II were determined using Western blot analysis in HaCaT keratinocytes treated with 0.25, 0.5 and 1 μg/mL PSMα3 for 0.5, 2, 6 and 24 h. The control HaCaT cells displayed endogenous expression of both the lipidated and non-lipidated forms of LC3B (Appendix A, lanes 1, 5, 9 and 13). PSMα3 increased the LC3B-II/LC3B-I ratios at all time points and for all PSMα3 concentrations (Appendix A, lanes 2–4, 6–8, 10–12 and 14–16). This result indicates that PSMα3 enhances the conversion of LC3B-I to the PE-conjugated LC3B-II. As PSMα3 used in a 1 μg/mL concentration triggered the highest increase in the LC3B-II/LC3B-I ratio at the 6 h time point, these conditions were chosen for our further experiments.

To investigate the autophagic flux in PSMα3-treated cells, LC3B-II levels were measured under conditions where autophagosome degradation was blocked by bafilomycin A1 (BFLA), a compound known to disrupt the fusion of autophagosomes with lysosomes and block the acidification of the autolysosome. The cultures were incubated with PSMα3 for 2 h and then treated with BFLA for another 4 h period. A short BFLA treatment was used to avoid assay saturation. The results show that PSMα3 elicited a significant increase in the level of LC3B-II (the fold increases in LC3B-II levels in cells treated with 0.5 and 1.0 μg/mL PSMα3 were 6.85 and 12.63, *p* < 0.05 and *p* < 0.001, respectively) (Figure 1A,B). These data indicate that PSMα3 increases the lipidation of LC3B-I. These experiments also reveal that BFLA increased the level of LC3B-II but to a greater extent in PSMα3-treated cultures than in the untreated cells (the fold increases in LC3B-II levels in cells treated with 0.5 and 1.0 μg/mL PSMα3 in the presence of BFLA were 45.61 and 42.19 vs. 28.42 in the BFLA control, *p* < 0.01 and *p* < 0.05, respectively) (Figure 1B). We used these data to gain insight into the autophagosome turnover in PSMα3-treated cells and analyzed formation, degradation and net turnover ratios separately (Figure 1C). The results show that 0.5 and 1.0 μg/mL PSMα3 significantly increased the autophagosome formation ratios (the fold increases in cells treated with 0.5 and 1.0 μg/mL PSMα3 were 1.63 and 1.51, *p* < 0.01 and *p* < 0.05, respectively) (Figure 1C). The autophagosome degradation ratios were higher for all toxin concentrations compared to that of the control, but these differences were not statistically significant (Figure 1C). Moreover, these data reveal that 1.0 μg/mL PSMα3 triggered a significant decrease in the net autophagic turnover rate (the fold decrease in cells treated with 1.0 μg/mL PSMα3 was 0.71, *p* < 0.01) (Figure 1C). These results demonstrate that 0.5 and 1.0 μg/mL PSMα3 stimulated autophagosome formation. Cells exposed to 0.5 μg/mL exotoxin can maintain the net turnover ratio at equilibrium, whereas 1.0 μg/mL PSMα3 triggers an overdriven, unbalanced autophagic activity.

### 3.2. The Effect of PSMα3 on the Subcellular Localization of LC3B and Beclin-1

To study the effect of 1.0 μg/mL PSMα3 on the intracellular localization of endogenous LC3B at the 6 h time point, an indirect immunofluorescence assay was used. The control cells displayed a pattern characterized by diffuse, faint, punctate and intense LC3B staining, and a few low-height peaks were visible in the 3D surface plots (Figure 2A). In contrast, the PSMα3-treated cells exhibited a pattern characterized by intense punctate LC3B staining, and numerous high peaks were visible in the 3D surface plots (Figure 2A). The PSMα3-treated cultures had a significantly greater average number of LC3B-positive vacuoles per cell than that of the control (the average numbers of autophagosomes in the control and PSMα3-treated cultures were 24.8 and 28.9, respectively, *p* < 0.01) (Figure 2B). These results indicate that PSMα3 stimulates the redistribution of LC3B from the cytoplasm to autophagosomes.

An indirect immunofluorescence assay was used to determine the effect of 1 μg/mL PSMα3 on the level and intracellular localization of Beclin-1 at the 0.5 h time point. Control cells displayed a pattern characterized by faint cytoplasmic Beclin-1 staining, and a few low-height peaks were visible in the 3D surface plots (Figure 3A). In contrast, the PSMα3-treated cells exhibited a pattern characterized by intense punctate Beclin-1 staining, and numerous high peaks were visible in the 3D surface plots (Figure 3A). PSMα3 significantly increased the Beclin-1 fluorescent signal intensities compared to the control (the CTCF value in PSMα3-treated cells was 1.61 vs. 1.0 in the control, *p* < 0.0001) (Figure 3B). Likewise, PSMα3 significantly increased Beclin-1-positive vacuole abundance compared to the control (the average number of Beclin-1-positive puncta in PSMα3-treated cells was 40.3 vs. 12.3 in the control group, *p* < 0.0001) (Figure 3C). These data indicate that PSMα3 elevates the level of Beclin-1. 

### 3.3. The Effect of PSMα3 on the Ultrastructural Features of the Autophagic Compartments

To investigate the ultra-structural features of autophagic compartments, transmission electron microscopy (TEM) was used. Normal cell morphology was found in the control cell cultures (Figure 4). In contrast, cells incubated with 1 μg/mL PSMα3 exhibited a considerable rise in the number of cytoplasmic vacuoles containing amorphous cell debris as early as 0.5 h after toxin treatment. The morphological characteristics of these organelles corresponded to autophagosomes and autolysosomes (Figure 4). At the 0.5 h time point, the average numbers of autophagosomes and autolysosomes per cytoplasmic µm^2^ in the control and PSMα3-treated cells were 0.3881 and 0.6316, respectively (*p* < 0.001) (Figure 4). Interestingly, we could observe the accumulation of early endosomes, multivesicular bodies/late endosomes and amphisomes composed of tubular and vacuolar parts 6 h after PSMα3 treatment (Figure 4). Amphisomes contained numerous (>6) intraluminal vesicles and damaged organelles (Figure 4). At the 6 h time point, the average numbers of amphisomes per unit of area in the control and PSMα3-treated cells were 0.03568 and 0.2768, respectively (*p* < 0.001) (Figure 4). At the 24 h time point, multilamellar bodies with a concentric pattern similar to onion leaves were visible in the cytoplasm of the cells incubated with PSMα3 (Figure 4). At the 6 h time point, the average numbers of multilamellar bodies per cytoplasmic µm^2^ in the control and PSMα3-treated cells were 0.008 and 0.221, respectively (*p* < 0.001) (Figure 4). These data are consistent with the electron microscopic picture of enhanced autophagy and confirm that PSMα3 stimulated the autophagic processes.

### 3.4. The Effect of PSMα3 on the Formation of Acidic Vesicular Organelles

To study the effect of 1.0 μg/mL PSMα3 on AVO formation, AO staining was used. Both control and PSMα3-treated cells displayed green nuclear and cytoplasmic staining (Figure 5A). AO staining revealed a few red puncta in the cytoplasm of control cells (Figure 5A). The average numbers of AVOs per cell in the controls were 16.2, 16.3 and 20.9 at 2, 6 and 24 h of culturing, respectively (Figure 5B). In the cytoplasm of PSMα3-treated cells, bright red puncta were observed; the average numbers of AVOs per cell were 27.8, 23.6 and 19.6 at the 2, 6 and 24 h time points, respectively (Figure 5A,B). This result demonstrates that PSMα3 significantly increased the average numbers of AVOs per cell at the 2 and 6 h time points. Thus, PSMα3 promotes cytoplasmic acidification and facilitates the development of AVOs. 

### 3.5. The Effect of PSMα3 on Some Cellular Signal Transduction Pathways Implicated in Autophagy Regulation

To determine the effect of 1.0 μg/mL PSMα3 on the phosphorylation levels of 37 protein kinases, a phospho-kinase array was used. PSMα3 led to the activation of extracellular signal-regulated kinases 1/2 (Erk1/2) by increasing the phosphorylation of residues T202/Y204 and T185/Y187 at the 0.5 h time point (Figure 6A,B). Compared with the control, the phosphorylation level of the proline-rich Akt substrate of 40 kDa (PRAS40) (T246) was also increased (Figure 6A,B). In contrast, the phosphorylation levels of c-Jun (S63), Ak strain transforming factor 1/2/3 (Akt1/2/3) (T308 and S473), proto-oncogene tyrosine-protein kinase Src (Y419), ribosomal protein S6 kinase (p70S6K) (T389 and T421/S424) and signal transducer and activator of transcription (STAT) family members, such as STAT1 (Y701) and STAT2 (Y689), were decreased (Figure 6A,B). Thus, the phosphorylation patterns of signaling molecules differ in the PSMα3-treated and control cells. 

To examine the involvement of mTOR in the PSMα3-mediated induction of autophagy, the levels of mTOR and phospho-mTOR (S2448) were determined via Western blot analysis (Figure 6C–F). The phosphorylation of S2448 in the negative regulatory domain of mTOR was shown to correlate with overall higher levels of mTOR activity [53,54]. Control HaCaT cells displayed endogenous expression of both the phosphorylated and unphosphorylated forms of mTOR (Figure 6C, lanes 1, 3, and 5). The phospho-mTOR/mTOR ratio was, on average, 2.96 at the 0.5 h time point in the control cells (Figure 6F). PSMα3 induced moderate increases in mTOR levels (Figure 6C; lanes 2, 4 and 6, and Figure 5D) and triggered significant decreases in the level of phospho-mTOR (Figure 6C, lanes 2, 4 and 6, and Figure 6E); the phospho-mTOR/mTOR ratio was, on average, 1.24 at the 0.5 h time point (Figure 6F). Thus, PSMα3 impairs mTOR activity by decreasing its phosphorylation at S2448.

## 4. Discussion

Compelling evidence indicates that autophagy plays a critical role in physiological processes and the pathological conditions of the skin. This catabolic process is essential for epithelial differentiation, colonization resistance, the maintenance of barrier function and immune protection. The structural components, exotoxins and metabolites of skin commensal and highly virulent bacteria can exert profound effects on keratinocytes by modulating their autophagic activity. Thus, this study investigated how PSMα3, a PFT produced by both skin commensal and pathogenic staphylococcal species, affects autophagy in HaCaT keratinocytes.

The present study provides evidence that PSMα3 stimulates autophagy. We evaluated the (i) levels of LC3B, (ii) autophagic flux, (iii) subcellular localization of LC3B and Beclin-1, (iv) ultra-structural features of cells and (v) cytoplasmic acidification in PSMα3-treated HaCaT cells. We first measured the levels of the autophagy marker protein, LC3B, using Western blot analysis. PSMα3 elevated LC3B-II and decreased LC3B-I levels at the 0.5, 2, 6 and 24 h time points (Appendix A), indicating that the lipidation of LC3B is stimulated by this staphylococcal PFT. We also assessed the autophagic flux by measuring the LC3B-II levels in control and PSMα3-treated cultures incubated with the vacuolar (H+)-ATPase inhibitor BFLA. In the presence of BFLA, the LC3B-II levels of PSMα3-treated cells were significantly higher than that seen in the drug control (Figure 1A,B), demonstrating that autophagic flux is increased by this toxin. To better understand the effect of PSMα3 on autophagosome turnover, we calculated the autophagosome formation, degradation and net turnover ratios. PSMα3, in 0.5 and 1 μg/mL concentrations, significantly increased the formation rates, and the degradation rates did not change (Figure 1C). The cells exposed to 0.5 μg/mL PSMα3 could maintain a steady-state net turnover rate (Figure 1C). This toxin, however, in a 1 μg/mL concentration, decreased the net turnover ratio significantly, reflecting a different relative rate of autophagosome formation and degradation (Figure 1C). These data demonstrate that PSMα3 stimulates autophagosome formation and can increase the autophagosome pool. Next, we investigated the intracellular distributions of LC3B and Beclin-1 using confocal microscopy. These experiments revealed that PSMα3 increases the intensity levels of LC3B and Beclin-1 and triggers the redistribution of these proteins from the cytoplasm to autophagosomes (Figure 2 and Figure 3). Furthermore, we used AO staining to determine the effect of PSMα3 on AVO formation. The results show that PSMα3 increased cytoplasmic acidification and promoted AVO formation (Figure 5). Finally, we studied the ultra-structural features of cells treated with PSMα3 using TEM. Interestingly, we could observe the accumulation of autophagosomes/autolysosomes, amphisomes and multilamellar bodies in cells treated with PSMα3 for 0.5, 6 and 24 h, respectively (Figure 4). Thus, our experiments demonstrate that sub-lytic concentrations of PSMα3 can stimulate autophagy in HaCaT keratinocytes. 

Autophagy has already been associated with cellular responses to various PFTs, such as the α-toxin (*S. aureus*), streptolysin O (*Streptococcus pyogenes*), listeriolysin O (*Listeria monocytogenes*) or *Vibrio cholerae* cytolysin (VCC) [47,48,55,56,57]. However, the mechanism and biological consequences of the pro-autophagic effect of bacterial PFTs are not fully understood. It has already been revealed that PFTs trigger pore formation, plasma membrane rearrangements, and deformations, leading to an influx of calcium and efflux of potassium [23,58]. Metabolic and ion imbalances activate some signaling pathways implicated in the demise or recovery of cells exposed to PFTs [23,59]. The staphylococcal α-toxin, streptolysin O, listeriolysin O and aerolysin, produced by *Aeromonas hydrophila*, activated the extracellular signal-regulated kinase (ERK) and p38 MAPK pathways that facilitate reconstitution of membrane integrity and ion homeostasis [60,61,62]. Other studies have revealed that aberrant stimulation of host protein phosphatases markedly attenuates Akt activation in cells exposed to staphylococcal α-toxin, aerolysin or α-hemolysin (HlyA) secreted by uropathogenic *Escherichia coli* [63]. Moreover, compelling evidence indicates that autophagy in cells exposed to PFTs contributes to maintaining metabolic homeostasis, restoring membrane integrity and promoting cellular recovery [23]. In light of these interesting observations, we investigated how PSMα3 affects the phosphorylation levels of protein kinases implicated in autophagy regulation. Our studies have shown that the level of phospho-Erk1/2 (T202/Y204 and T185/Y187) was increased in PSMα3-treated cells (Figure 6A,B). This result demonstrates that this staphylococcal PFT, like other PFTs, activates the cytoprotective ERK pathway. It has also been revealed that c-Jun and JunB inhibit autophagy [64]. Jun proteins can form homo- and heterodimers with c-Fos, ATF and Maf subfamily members constituting activator protein-1 (AP1). The AP1 transcription factor stimulates the expression of several genes, including Bcl-2, Bcl-xL and Beclin-1 ([65]). The transcriptional activity of Jun proteins was shown to be required for their pro-autophagic effect [64]. The altered Beclin-1/Bcl2 ratio leads to a pro-autophagic balance and enhanced autophagic activity [25]. Our experiments show that the level of phospho-c-Jun (S63) was decreased in PSMα3-treated cells (Figure 6A,B). In light of the previous observations, it is reasonable to infer that the PSMα3-mediated suppression of phospho-c-Jun can also contribute to the induction of autophagy. In contrast, the levels of phospho-mTOR (S2448), phospho-Akt1/2/3 (T308 and S473), phospho-STAT1 (Y701) and phospho-STAT2 (Y689) were decreased in response to PSMα3 (Figure 5A,B). Under normal metabolic conditions, active mTORC1 inhibits the catabolic autophagic process. The class I phosphatidylinositol 3-kinase (PI3K)/Akt pathway is known to stimulate mTORC1 via phosphorylation events. Akt activates mTORC1 by phosphorylating tuberous sclerosis 2 protein at multiple sites and PRAS40 at Thr246 [25,66]. In contrast, harmful environmental cues trigger mTORC1 inactivation and thereby stimulate autophagy. The decreased phosphorylation levels of Akt1/2/3, mTOR and p70S6K indicate that the deactivation of the Akt/mTOR/p70S6K pathway is implicated in the mechanism of the PSMα3-mediated induction of autophagy. 

Previous studies have demonstrated that the autophagic pathway and the endolysosomal system interact closely. Autophagosomes and multivesicular bodies/late endosomes can fuse to generate amphisomes, which can eventually fuse with lysosomes or mature into multilamellar bodies [67,68,69,70]. Previous interesting studies have also demonstrated that streptolysin O pores are removed from the plasma membrane by endocytosis [71]. Toxin-carrying endocytic vesicles traffic along the endosomal pathway to multivesicular bodies, which eventually fuse with lysosomes, leading to the destruction of toxin molecules [71]. A similar mechanism for removing VCC pores was reported [72]. This study also showed that autophagy plays a role in the formation of multivesicular bodies, promotes the degradation of VCC pores and confers resistance to destructive toxicity [72]. Our TEM experiments show that, initially, autophagosomes/autolysosomes; later, amphisomes; and finally, multilamellar bodies accumulate in PSMα3-treated HaCaT keratinocytes (Figure 4). The proportions of the autophagosomes/autolysosomes, amphisomes and multilamellar bodies varied significantly within the cytoplasmic vacuole pools over time (Figure 4). Based on these observations, we suggest that PSMα3 enhances the intensive and concerted action of the endocytic and autophagic pathways, thereby facilitating the degradation or sequestration of toxin molecules and cellular survival of keratinocytes. The limitation of the present study includes the lack of data obtained by using primary keratinocytes and several other cell types. 

Thus, our study provides evidence that PSMα3 exerts a pro-autophagic effect in an in vitro cultured keratinocyte cell line.

## Figures and Tables

**Figure 1 biomedicines-11-03018-f001:**
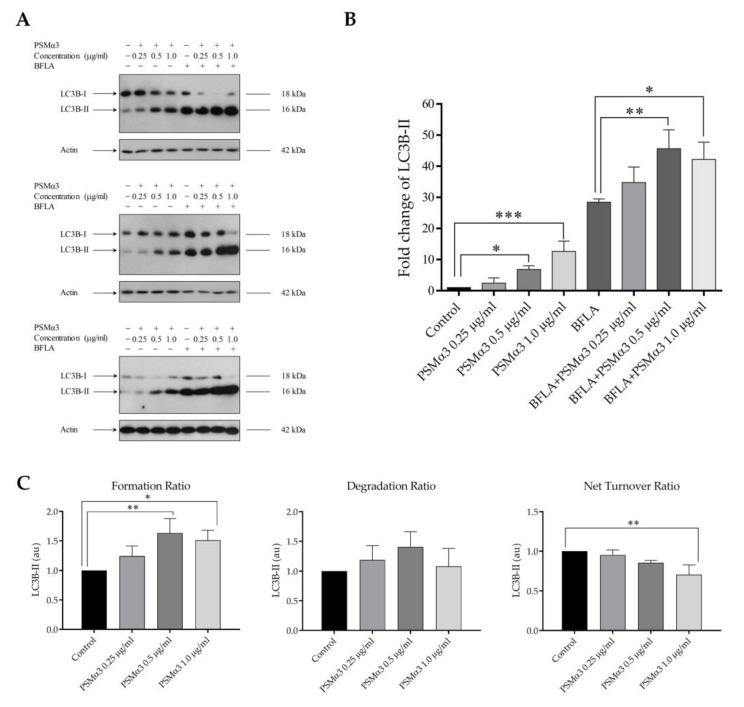
PSMα3 stimulates autophagic flux. After being exposed to 0.25, 0.5 and 1 µg/mL PSMα3 for 2 h, HaCaT cells were treated with 100 nM BFLA for an additional 4 h. To determine LC3B-I and LC3B-II levels, Western blot analysis was used. (**A**) Western blot analysis. The results of three independent experiments are shown. (**B**) The ratios of the protein levels measured in the cells treated with PSMα3 and BFLA alone or in combination were compared to that in the control and expressed as fold change. The values on the bar graph denote the means ± SD of the results of three independent experiments. (**C**) The effect of PSMα3 on autophagosome turnover. The graphs show the autophagosome formation, degradation and net turnover ratios. *p* values were calculated with an ANOVA with the Sidak post hoc test. * *p* < 0.05, ** *p* < 0.01, *** *p* < 0.001.

**Figure 2 biomedicines-11-03018-f002:**
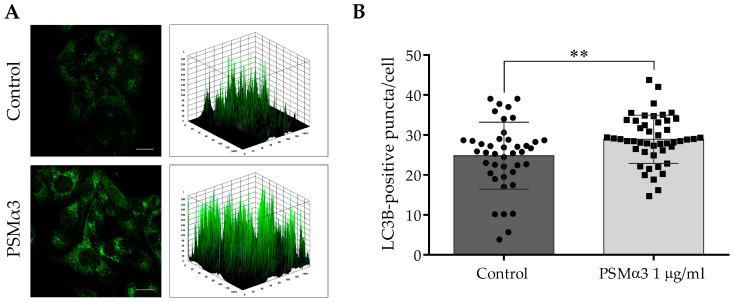
PSMα3 stimulates autophagosome formation. HaCaT cells were treated with 1 µg/mL PSMα3 for 6 h. Control cultures were left untreated for comparison. (**A**) Immunofluorescence assays showing the fluorescence intensities of LC3B-positive vacuoles. The samples were stained for endogenous LC3B protein, and confocal microscopy was used to capture images, which were subsequently subjected to fluorescence intensity analysis. The data represent two independent experiments. (**B**) The average numbers of LC3B-positive autophagic vacuoles. The means ± standard deviation (SD) of the results are depicted on the bar graph. Statistical analysis was performed using an unpaired *t*-test. **, *p* < 0.01. Scale bars, 20 μm.

**Figure 3 biomedicines-11-03018-f003:**
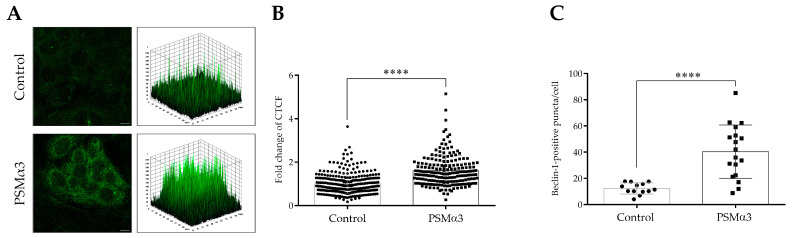
PSMα3 increases the level of Beclin-1. HaCaT cells were treated with 1 µg/mL PSMα3 for 0.5 h. Control cultures were left untreated for comparison. (**A**) Immunofluorescence assays showing the fluorescence intensities of Beclin-1. Confocal microscopy was used to capture images, which were subsequently subjected to fluorescence intensity analysis. The results are representative of two independent experiments. (**B**) Quantification of Beclin-1 fluorescence intensities. The fold changes in CTCF values were calculated as the CTCF of cells treated with 1 µg/mL PSMalha3/CTCF of control cultures. The values on the bar graph denote the means ± SD. (**C**) The average numbers of Beclin-1-positive puncta. The intracellular abundances of Beclin-1-positive puncta were quantified using ImageJ software. The values on the bar graph denote the means ± SD. *p* values were calculated with an unpaired *t*-test. ****, *p* < 0.0001. Scale bars, 20 µm.

**Figure 4 biomedicines-11-03018-f004:**
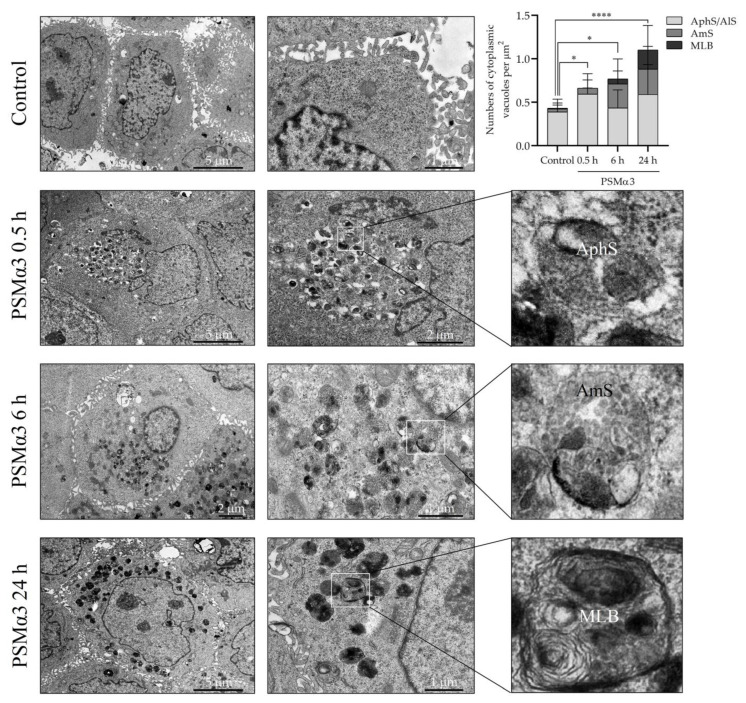
PSMα3 alters the ultra-structural characteristics of the autophagic compartments of HaCaT keratinocytes. HaCaT cells were treated with 1 μg/mL PSMα3 for 0.5, 6 and 24 h, and the autophagic compartment profiles were evaluated via TEM. The solid line boxes encompassing cytoplasmic portions of cells were further enlarged in the inserts to show autophagosomes, amphisomes and multilamellar bodies. AphS, autophagosome; AmS, amphisome; MLB, multilamellar body. *, *p* < 0.05, ****, *p* < 0.0001.

**Figure 5 biomedicines-11-03018-f005:**
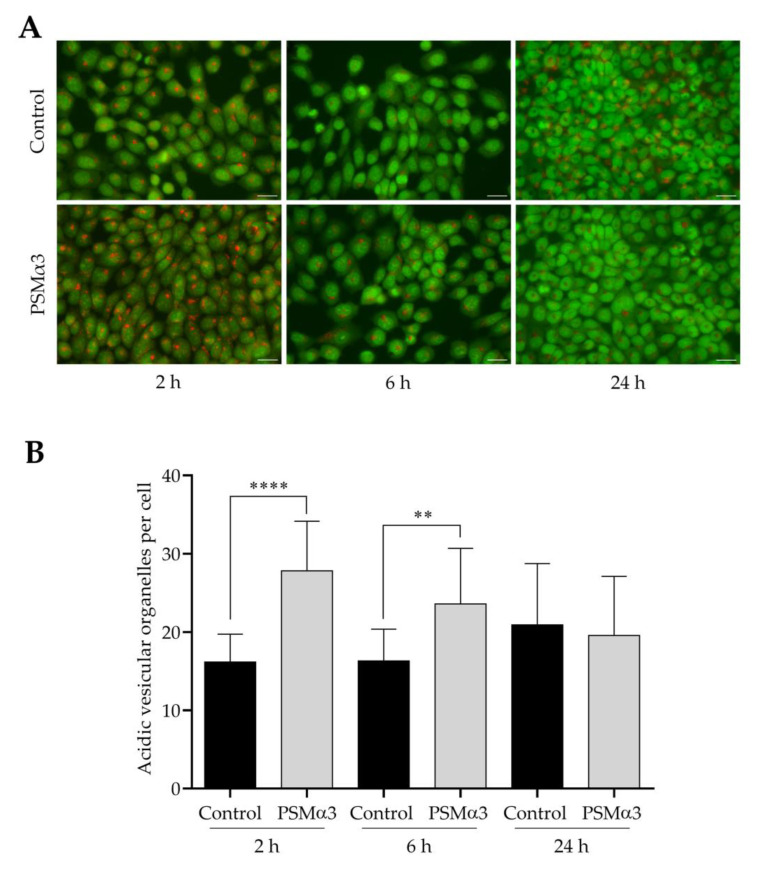
PSMα3 stimulates AVO formation. (**A**) Representative fluorescence micrographs showing the intracellular localization of AVOs. Control cells and cultures treated with 1 μg/mL PSMα3 for 2, 6 and 24 h were stained with AO. After the cells had been washed to remove excess AO, images were obtained with a Nikon ECLIPSE Ts2 Inverted LED Phase Contrast Microscope. (**B**) Quantification of AVOs. The average numbers of AVOs per cell were determined by using Image J software. The values on the bar graph denote the means ± SD. *p* values were calculated with an unpaired *t*-test. **, *p* < 0.01; ****, *p* < 0.0001. Scale bars, 20 μm.

**Figure 6 biomedicines-11-03018-f006:**
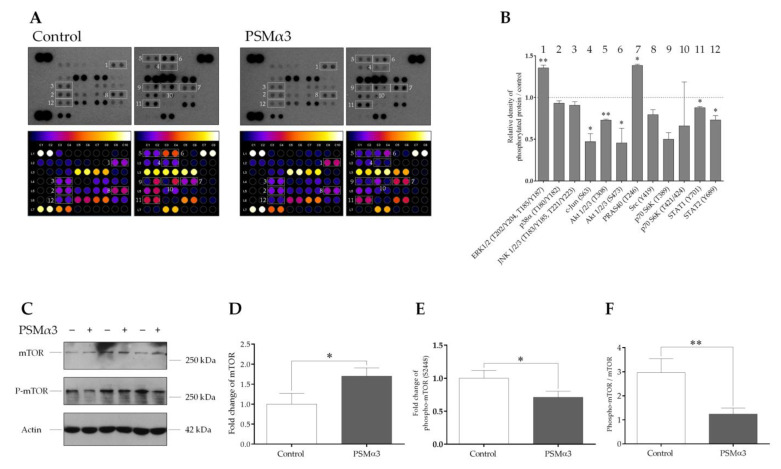
PSMα3 alters the phospho-kinase array profile of HaCaT keratinocytes. (**A**) Phospho-kinase array analysis. HaCaT cells were treated with 1 μg/mL PSMα3 for 0.5 h, whereas control cells were left untreated. Total protein was isolated and subjected to phospho-kinase array analysis. The labeled spots correspond to the phospho-proteins modulated by PSMα3. (**B**) Quantification of phosphoproteins from the proteomic array. Spot densities of phosphoproteins were quantified using ImageJ analysis software and normalized to positive controls on the same membrane. (**C**–**F**) PSMα3 decreases phosphorylation of mTOR at Ser2448. Total protein was isolated from control and PSMα3-treated cells and were analyzed for mTOR and phospho-mTOR (S2448) expression via Western blot analysis. Results are representative of three independent experiments. The values on the bar graphs denote means ± SD. *p* values were calculated with an unpaired *t*-test. *, *p* < 0.05; **, *p* < 0.01.

## Data Availability

All data generated and analyzed during this study are included in this manuscript and its Appendix A.

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
