# Peer review of "Phenol-Soluble Modulin α3 Stimulates Autophagy in HaCaT Keratinocytes"

_biomedicines, 2023, doi:10.3390/biomedicines11113018_

Round 1

Reviewer 1 Report

Comments and Suggestions for Authors

General Comments

This study demonstrates an inducible effect of PSMα3 on autophagy process in HaCaT keratinocytes.This information is very important and interesting and may help us understanding the molecular pathway and the target molecule in this regards. However, major revision is required. Please revise your manuscript.

Major Points

The study is well conducted, the experiments are clearly explained and analysed.The study brings interesting and novel conclusions to the field.
 A major concern stays in the fact that the authors only analysed one keratinocyte cell line (HaCaT), which is spontaneously immortalized. In order to confirm their results, they have to reproduce their observations (at least figure 1 to 6, except figure 3) in two additional primary human keratinocyte cell lines that should be derived from different individuals. Moreover, it would be really interesting to compare also the results from primary human keratinocytes to results with the HaCaT keratinocyte cell line.

1. Title page

Titel: I would suggest to add in human keratinocytes to Title:  Phenol-Soluble Modulinα3 Stimulates Autophagy in Human Keratinocytes.

2. Abstract

“To study the effect of PSMα3 in keratinocytes, exert diverse cellular effect…”: the authors should explain the reason why they have treated primary HaCaT with PSMα3 and investigate autophagy process? Did they look at apoptosis phenomena?

3. Introduction

Page 4 line 130 Intracellular infection of many cell types…please give examples of cell types with references.

4. Material and Methods

The authors should write the used solution of autophagy inhibitor bafilomycin A1 (BFLA)(Page 4, line 150) and if they used DMSO, is there any experiment regarding western blot or indirect immunofluorescence with DMSO alone as control?

5. Results

Page 6, lines 259-261, please check the results regarding expression of LC3B-I and LCB-II, as the authors interpret other results in this regard in the discussion section (Page 12, Line 454).

It is confusing that in your experiment (Figure 1A and 1B, page 7) regarding western blot analysis the expression of LC3B-II in combination with PSMα3 and BFLA is even higher compared to cells treated with PSMα3 alone. Since BFLA is actually supposed to block autophagy, they do not expect an increase in the expression of LC3B-II in your model system, please clearly state the reason.

In Figure 2 which type of LC3B proteins are stained, LC3B-I or LC3B-II?

In Figure 3, how many times did the authors examine by TEM analysis? And the analysis was done by treatment with PSMα3 but not by BFLA.Why was the inhibitor of autophagy BFLA not checked in electron microscopy examination?

Please add scale bar in the text in Figure 3.

In your study the authors first pretreated the HaCaT keratinocyte cells with autophagy stimulant PSMα3 and then the autophagy inhibitor BFLA was given. Why wasn't inhibitor taken first? Please clearly state the reason in the paper.

Comments on the Quality of English Language

Please review the text in the paper by a native English speaker.

Author Response

Referee 1

We are grateful for the Referee for the reviewing work.

”The study is well conducted, the experiments are clearly explained and analysed.The study brings interesting and novel conclusions to the field. A major concern stays in the fact that the authors only analysed one keratinocyte cell line (HaCaT), which is spontaneously immortalized. In order to confirm their results, they have to reproduce their observations (at least figure 1 to 6, except figure 3) in two additional primary human keratinocyte cell lines that should be derived from different individuals. Moreover, it would be really interesting to compare also the results from primary human keratinocytes to results with the HaCaT keratinocyte cell line.”

We fully agree with the Referee on the importance of investigating the pro-autophagic effect of PSMs in multiple cell lines, especially primary human keratinocyte cell lines. We have chosen the HaCaT cell line to ensure the required cell amount and minimize the inter-experimental variability. We did not plan to use primary keratinocytes at this stage of our research work. We intend to pursue this direction in future research when we have access to the necessary resources and primary cell cultures. Obtaining the ethical board approval required for studies involving human specimens would take several months. Although we can not include data obtained with primary human keratinocytes, we have modified the title (see lines 2-3) and described the limitations of our study in the Discussion section (see lines 618-620).  

“1. Title page Titel: I would suggest to add in human keratinocytes to Title:  Phenol-Soluble Modulinα3 Stimulates Autophagy in Human Keratinocytes.”

We have modified the title (see lines 2-3).

 “2. Abstract: “To study the effect of PSMα3 in keratinocytes, exert diverse cellular effect…”: the authors should explain the reason why they have treated primary HaCaT with PSMα3 and investigate autophagy process? Did they look at apoptosis phenomena?”

The Abstract has reached the upper limit specified in the Authors' instructions, thus it cannot be further expanded.

PSMα was shown to induce necroptotic cell demise through RIP1/RIP3/MLKL signaling in the THP-1 cell line and human primary macrophages (Kitur K, Parker D, Nieto P, Ahn DS, Cohen TS, Chung S, Wachtel S, Bueno S, Prince A. Toxin-induced necroptosis is a major mechanism of Staphylococcus aureus lung damage. PLoS Pathog. 2015 Apr 16;11(4):e1004820. doi: 10.1371/journal.ppat.1004820. PMID: 25880560.). It has also been demonstrated that PSMα1, 2, and 3 trigger necroptosis by stimulating MLKL phosphorylation in murine neutrophil granulocytes (Zhou Y, Niu C, Ma B, Xue X, Li Z, Chen Z, Li F, Zhou S, Luo X, Hou Z. Inhibiting PSMα-induced neutrophil necroptosis protects mice with MRSA pneumonia by blocking the agr system. Cell Death Dis. 2018 Mar 2;9(3):362. doi: 10.1038/s41419-018-0398-z. PMID: 29500427.). Necroptosis induced by PSMs results in cell lysis, the mechanism of which differs from that resulting from the disintegration of the plasma membrane. These data indicate that PSMs cause cell death by triggering plasma membrane disintegration and necroptosis. Thus, apoptosis was not studied in our experiments.

“3. Introduction: Page 4 line 130 Intracellular infection of many cell types…please give examples of cell types with references.”

Examples of cell types have now been included in the Introduction section (see lines 136-137 and the Reference list).

“4. Material and Methods: The authors should write the used solution of autophagy inhibitor bafilomycin A1 (BFLA)(Page 4, line 150) and if they used DMSO, is there any experiment regarding western blot or indirect immunofluorescence with DMSO alone as control?”

The solvent of BFLA has now been given in the Materials and Methods (see line 163). In our experiments, BFLA was used to measure the effect of PSMα3 on the autophagic flux. We used 0.05% DMSO concentration only for 4 hours. We verified that DMSO does not significantly affect the autophagic activity of the cells when such parameters are used (data not shown).

“5. Results: Page 6, lines 259-261, please check the results regarding expression of LC3B-I and LCB-II, as the authors interpret other results in this regard in the discussion section (Page 12, Line 454).”

We have determined the kinetics of LC3B-II expression in cells treated with PSMα3 for 0.5, 2, 6 and 24 h, and calculated the LC3B-II/LC3B-I ratios. These results are shown in Supplementary Figure 1. These data are interpreted in the Discussion section.

“It is confusing that in your experiment (Figure 1A and 1B, page 7) regarding western blot analysis the expression of LC3B-II in combination with PSMα3 and BFLA is even higher compared to cells treated with PSMα3 alone. Since BFLA is actually supposed to block autophagy, they do not expect an increase in the expression of LC3B-II in your model system, please clearly state the reason.”

BFLA is a pharmacological inhibitor of autophagosome-lysosome fusion and lysosomal hydrolase activity. In cells incubated with BFLA lysosomal degradation of LC3B-II is inhibited. Thus, the level of LC3B-II is expected to be increased.

“In Figure 2 which type of LC3B proteins are stained, LC3B-I or LC3B-II?”

The only difference between LC3B-I and LC3B-II is that LC3B-II is conjugated with phosphatidylethanolamine. The antibody preparation is protein-specific and recognizes both LC3B-I and LC3B-II.  The lipidation alters the electrophoretic mobility of LC3B-II. That is why the two proteins can be detected by Western blot analysis separately.

“In Figure 3, how many times did the authors examine by TEM analysis? And the analysis was done by treatment with PSMα3 but not by BFLA. Why was the inhibitor of autophagy BFLA not checked in electron microscopy examination?”

During our TEM analysis, we took many pictures at various magnifications (2000-, 2500-, 3000-, 5000-, 6000-, 8000-, 10000-, 12000-, 15000-, 20000-, and 25000-fold). Now we counted the autophagosomes/autolysosomes, amphisomes and multilamellar bodies in a cytoplasmic area of 1500-2000 um2 per sample. These new data show that, there is a drift from autophagosomes/autolysosomes through amphisomes to multilamellar bodies. Accordingly, we have modified the Discussion section (lines 655-659). We have also included a new graph (see Figure 4), modified the Results (see lines 408-411, 413-416 and 418-420), and Materials and Methods (see lines 275-280). We are grateful to the Referee for this comment. BFLA is recommended for the investigation of the autophagic flux. For the measurement of flux, western blot analysis is recommended. TEM is not suitable for examining the autophagic flux, due to the extremely large number of autophagosomes accumulating in the cells.

“Please add scale bar in the text in Figure 3.”

The scale bars have now been added (see Figure 4).

“In your study the authors first pretreated the HaCaT keratinocyte cells with autophagy stimulant PSMα3 and then the autophagy inhibitor BFLA was given. Why wasn't inhibitor taken first? Please clearly state the reason in the paper.”

According to the autophagy guidelines, BFLA treatment needs to be relatively short (1–4 h) to allow comparisons of the amount of LC3 that is lysosomally degraded over a given time frame under one treatment condition to another treatment condition. Keeping BFLA treatment short helps to avoid assay saturation. We wished to measure autophagic flux in cells treated with PSMα3 for 6 h and BFLA for 4 h. Therefore, PSMα3 was added to the cells 2 h earlier than BFLA. Accordingly, we have changed the results section (see lines 307-308).  

We are grateful for the Referee for the reviewing work and consider that the modifications really do improve the manuscript.

Reviewer 2 Report

Comments and Suggestions for Authors

Make corrections as the attached file 

Comments on the Quality of English Language

Minor editing of the English language required

Author Response

Dear Referee 2,

We sincerely appreciate the time and effort you dedicated to reviewing our manuscript, "Phenol-Soluble Modulin α3 Stimulates Autophagy." Your feedback and constructive suggestions have been invaluable to us, and we are grateful for your thoughtful assessment of our work.

Sincerely yours,

                                                                                   Klára Megyeri M.D., Ph.D.

                                                                                       corresponding author

Reviewer 3 Report

Comments and Suggestions for Authors

The manuscript describes the effect of the staphylococcal virulent factor PSMα3 on autophagy in the HaCaT keratinocyte cell line. The authors show that PSMα3 increases LC3B-II expression levels with a concomitant decrease in LC3B-I expression levels and from this they calculated the formation and degradation ratio as well as the net turnover ratio. They further showed increased LC3B-stained vacuoles, and TEM images show a gradual development of multilamellar bodies via autophagosomes/autolysosomes and amphisomes. Visually the autophagosomes or autolysosomes appear as large vacuoles much larger than what is expected for autophagosomes or autolysosomes (ranging from a few hundred nanometers to several micrometers). The authors further showed that there is an increase in acridine orange-stained acidic vesicular organelles. Then they studied some changes in the phosphorylation of components of different signal transduction pathways, and at the end the expression level of Beclin-1. I think the latter should be shown together with the expression of LC3B-I and LC3B-II. In general, the manuscript is well written. There are some issues that should be addressed before the manuscript can be accepted for publication.

Line 26: Please remove "their"

In line 22 their mention "pro-autophagic" while in the next sentence they state that it has not been studied. So please rephrase.

There is a paper describing modulation of autophagy by S. aureus that was dependent on high Agr expression, which regulates PMS expression (https://doi.org/ 10.1128/iai.00358-15). This paper should be included in the manuscript.

Lines 26, 30 and all other places: Capital letter in Western.

Line 33: Is the apparent increased autophagy flux due to the decreased autophagic turnover rate?

Line 35: When you use the word "respectively", do you intend that each of the autophagy-related structures were accumulated in the respective time point, meaning a drift from autolysosomes to amphisomes to multilamellar bodies? If so, it means an interference in downstream processing. Thus, PSMα3 does more than just stimulating autophagy, and this should appear in the title.

Line 38 The conclusion is not clear – it is more like a far-fetched interpretation. Suddenly the virulence factor PSMα3 becomes a protective factor on skin homeostasis. PSMα3 is produced by the bacteria and not by the cellular defense mechanisms, so it can not be an integral part of the cellular defense mechanism.

Line 53: S. epidermidis should be in italics.

Line 62: CoPS should be defined first time used.

Lines 100 and 138 and other places: Staphylococci should be written with capital S and be in italics.

Line 102: PFT should be defined first time used.

By the way the line spacing of Introduction should be 1.0 as in the other sections.

Lines 122-123: The difference between LC3B-I and LC3B-II should be described.

Lines 131-133: The sentence is not clear. It says that S. aureus are destroyed by pyroptosis, but it is S. aureus that kill the host cells by pyroptosis. Please rephrase the sentence.

The host cell types tested in papers mentioned in Introduction should be stated.

Line 158: Was the FCS heat-inactivated? If so, this should be stated.

Line 162: "2" should be in subscript (CO2).

Line 163: The preparation of cytospin samples should be described.

Section 2.4 – The color and fluorescence filters should be added to the text.

Section 2.5: The catalog number of all primary and secondary antibodies should be provided, and the name of the imaging system. The percentage of gel and instrument used to separate LC3B-I and LC3B-II should be described in the text.

The catalog number of the human phospho-kinase array should be provided.

Line 217: What do you mean with 100-100?

Section 2.6: The concentrations of each of the antibodies should be stated.

Line 230: What was the size of the Dextran, and the source? Please add catalog number.

Line 230: Correct to OsO4.

Line 233: Add the concentration and time of incubation with the compounds.

Line 250: Describe what is the meaning of a higher LC3B-II/LC3B-I ratio.

Line 252: What are the levels of PSMs in fluid of Staphylococci infections? Is 1 μg/ml close or far from these values?

Line 258: Correct to 4 h. The same for line 304: 6 h, and other places remove the hyphen between the number and h.

Line 286: What do you mean with: "which does not seem sustainable in the long term"

Legend to Figure 1B: The calculations against actin should be described.

Figure 1C: The (au) in Y-axis should be replaced with a text of what was measured. The calculation of formation, degradation and net turnover ratio should be described in the Method section.

Line 322: The procedure of preparing the 3D surface plots should be described in the Method section.

Line 325: Describe in the Method section how the LC3B-positive autophagic vacuoles were automatically quantified by using the Image J 325 software.

Figure 2 should show the staining also after 0.5, 2 and 24 h, and concomitant bright light images of the cells, to see if these undergo similar changes as shown by TEM.

How does PSMα3 affect the metabolic activity and viability of the cells?

Figure 3: Is the large vacuolization indeed due to autophagy? What happens if you add an autophagy inhibitor? The transformation of vacuoles to amphisomes and MLB, suggests that some steps of the autophagy process is interrupt. If it is only stimulation of autophagy, we should see more degradations, but here we have an accumulation of membranes. Can this process be quantified. The images are from one cell. More images are required. It is recommended to perform a time-lapse confocal microscopy using Nile red or another membrane stain to show the entire process and to follow the intracellular fate of the autolysosomes. This would really provide an information about what is going on, and how fast the large autolysosomes are formed. The size of the bars should be shown, and bars of the highest magnification should also be shown. What is the normal size of autolysosomes? How can you know that these are autolysomes and not autophagosomes?

Line 356: AVO should be defined first time used.

Interestingly is the reduced phosphorylation of c-Jun that is also involved in regulating autophagy. Please comment on it.

Line 417: Remove hyphen.

Line 478: Please mention in the manuscript, at which concentrations of PSMα3 are lytic.

Comments on the Quality of English Language

The English is in general fine with minor English editing required.

Author Response

Dear Referee 3,

We are grateful for the Referee for the reviewing work.

Then they studied some changes in the phosphorylation of components of different signal transduction pathways, and at the end the expression level of Beclin-1. I think the latter should be shown together with the expression of LC3B-I and LC3B-II.”

The section describing the effect of PSMα3 on the level of Beclin-1 has been removed as requested by the Referee, and now it is shown together with the expression of LC3B-I and LC3B-II (see lines 376-400).

Line 26: Please remove "their"

”their” has been removed (see line 23).

In line 22 their mention "pro-autophagic" while in the next sentence they state that it

has not been studied. So please rephrase.”

We are grateful to the Referee, the effect of PSMα3 on autophagy has not really been studied so far. The word ”pro-autophagic” has therefore been deleted (see lines 23-24).

There is a paper describing modulation of autophagy by S. aureus that was dependent on high Agr expression, which regulates PMS expression (https://doi.org/10.1128/iai.00358-15). This paper should be included in the manuscript.”

This very interesting and relevant paper has now been included in the manuscript (see lines 103-106, and the Reference list).

Lines 26, 30 and all other places: Capital letter in Western.”

The writing of Western has now been corrected (see line 27 and all other places).

Line 33: Is the apparent increased autophagy flux due to the decreased autophagic turnover rate?”

The net autophagic turnover rate is determined by the relative speed of autophagosome formation and degradation. In cells treated with PSMα3, the rate of autophagosome formation is higher than that of the degradation. Therefore, the net turnover ratio (Degradation Ratio/Formation Ratio) decreases, and the size of the autophagosome pool increases. There are two possibities why formation and degradation of autophagosomes dissociate in response to PSMα3. One possibility is that PSMα3 affects autophagosome formation and lysosomal function in an uncoordinated manner, or the degradative capacity of lysosomes may become overloaded over time.

Line 35: When you use the word "respectively", do you intend that each of the autophagy-related structures were accumulated in the respective time point, meaning a drift from autolysosomes to amphisomes to multilamellar bodies? If so, it means an interference in downstream processing. Thus, PSMα3 does more than just stimulating autophagy, and this should appear in the title.”

Yes, that's exactly what we meant. As a result of this careful observation of the Referee, we have now performed the quantification of these cytoplasmic vacuoles. During our TEM analysis, we took many pictures at various magnifications (2000-, 2500-, 3000-, 5000-, 6000-, 8000-, 10000-, 12000-, 15000-, 20000-, and 25000-fold). Now we counted the autophagosomes/autolysosomes, amphisomes and multilamellar bodies in a cytoplasmic area of 1500-2000 um2 per sample. These new data show that indeed, there is a drift from autophagosomes/autolysosomes through amphisomes to multilamellar bodies. As we have not investigated this effect at the molecular level, we feel that further investigations would be necessary to change the title of the manuscript. However, to emphasize this phenomenon, we have modified the Discussion section (lines 655-659). We have also included a new graph (see Figure 4), modified the Results (see lines 408-411, 413-416 and 418-420), and Materials and Methods (see lines 275-280). We are grateful to the Referee for this comment.

“Line 38 The conclusion is not clear – it is more like a far-fetched interpretation. Suddenly the virulence factor PSMα3 becomes a protective factor on skin homeostasis. PSMα3 is produced by the bacteria and not by the cellular defense mechanisms, so it can not be an integral part of the cellular defense mechanism.”

To clarify this issue, we have modified the Abstract and Discussion (see lines 38 and 623-625).

“Line 53: S. epidermidis should be in italics.”

We, fully agree with the Referee, and it has now been corrected (see line 56).

“Line 62: CoPS should be defined first time used.”

We, again agree with the Referee, and CoPS has now been defined (see line 65).

“Lines 100 and 138 and other places: Staphylococci should be written with capital S and be in italics.”

According to the CDC style guide, the genus name in plural should not be capitalized or italicized (Corporate Authors(s): Centers for Disease Control and Prevention (U.S.), CDC style guide, 2023, p. 24.), thus, we would like to adhere to the spelling we used.

“Line 102: PFT should be defined first time used.”

Please see line 64.

“the line spacing of Introduction should be 1.0 as in the other sections.”

We are grateful to the Referee, the line spacing has now been corrected.

 “Lines 122-123: The difference between LC3B-I and LC3B-II should be described.”

We agree, the difference between LC3B-I and LC3B-II has now been described (see line 128).

“Lines 131-133: The sentence is not clear. It says that S. aureus are destroyed by pyroptosis, but it is S. aureus that kill the host cells by pyroptosis. Please rephrase the sentence.”

Although pyroptosis triggers demise of host cells, IL-1, released by dying cells, activates phagocytes, which destroy bacteria. Pyroptosis contributes to the elimination of S. aureus in an indirect way (Soong, G.; Paulino, F.; Wachtel, S.; et al. Methicillin-Resistant Staphylococcus Aureus Adaptation to Human Keratinocytes. mBio 2015, 6, 10.1128/mbio.00289-15, doi:10.1128/mbio.00289-15). The text has been modified (see lines 139-141).

“The host cell types tested in papers mentioned in Introduction should be stated.”

The cell types have now been included in the Introduction (see lines 136-137).

“Line 158: Was the FCS heat-inactivated? If so, this should be stated.”

Heat-inactivated FCS was used. This information has now been included in the manuscript (see line 170).

“Line 162: "2" should be in subscript (CO2).”

The mistake has now been corrected (see line 170).

“Line 163: The preparation of cytospin samples should be described.”

The preparation of cytospin samples has now been included in the manuscript (see lines 173-176).

“Section 2.4 – The color and fluorescence filters should be added to the text.”

The color and fluorescence filters have now been included in the manuscript (see lines 204-207).

“Section 2.5: The catalog number of all primary and secondary antibodies should be provided, and the name of the imaging system. The percentage of gel and instrument used to separate LC3B-I and LC3B-II should be described in the text. The catalog number of the human phospho-kinase array should be provided.”

The requested information has now been given in the manuscript (see lines 224-230, and 250-251).

“Line 217: What do you mean with 100-100?”

The confusing part of this sentence has been deleted (see line 255).

“Section 2.6: The concentrations of each of the antibodies should be stated.”

In the phosphokinase array kit, an antibody cocktail is used containing 37 different antibodies, the concentration of which might be different. Thus, the concentration of each of the antibody can not be given. According to the manufacturer’s instruction, the antibody cocktails and the streptavidin-peroxidase were used at 1:50 and 1:2000 dilutions, respectively. This information has now been given in the manuscript (see lines 260-262).

“Line 230: Correct to OsO4.”

OsO4 has now been corrected (see line 269).

“Line 233: Add the concentration and time of incubation with the compounds.”

The concentration and time of incubation with uranyl acetate and lead citrate have now been included in the manuscript (see lines 272-273).

“Line 250: Describe what is the meaning of a higher LC3B-II/LC3B-I ratio.”

We agree with the Referee, and we have included a new sentence to explain the meaning of the increased LC3B-II/LC3B-I ratios observed in PSMα3-treated cells (see lines 296-298).

“Line 252: What are the levels of PSMs in fluid of Staphylococci infections? Is 1 μg/ml close or far from these values?”

To our knowledge, the PSMα concentrations of clinical samples obtained from patients with staphylococcal infections have not yet been determined. However, it has previously been shown that clinical isolates of S. epidermidis and strains obtained from healthy individuals secrete PSMα. The mean concentrations of PSMα were 63.5 ± 16.6 mg/ml and 86.0 ± 27.0 mg/ml in the culture supernatants of clinical isolates and strains from healthy individuals, respectively (Vuong C, Dürr M, Carmody AB, Peschel A, Klebanoff SJ, Otto M. Regulated expression of pathogen-associated molecular pattern molecules in Staphylococcus epidermidis: quorum-sensing determines pro-inflammatory capacity and production of phenol-soluble modulins. Cell Microbiol. 2004 Aug;6(8):753-9. doi: 10.1111/j.1462-5822.2004.00401.x. PMID: 15236642.). Although the PSMα concentration measured in the culture supernatants is undoubtedly higher than that we used, we think that during infections, some cells can be exposed to a PSMα concentration of 1 μg/ml, as the density of live bacteria in certain tissues may be lower than in bacterial cultures. Moreover, in the case of bacteremia, we can expect a significant dilution of the toxin in the blood. Thus, it is reasonable to infer that, during infections, the toxin concentration we used can develop.

“Line 258: Correct to 4 h. The same for line 304: 6 h, and other places remove the hyphen between the number and h.”

Our language editor recommended the usage of hyphens, thus we would like to keep them.

“Line 286: What do you mean with: "which does not seem sustainable in the long term"”

This confusing remark has been deleted from the text of the manuscript (see lines 334-335).

“Legend to Figure 1B: The calculations against actin should be described.”

Western blot normalization has now been included in the Materials and Methods section (see lines 232-239).

“Figure 1C: The (au) in Y-axis should be replaced with a text of what was measured. The calculation of formation, degradation and net turnover ratio should be described in the Method section.”

Figure 1C has been corrected (see Figure 1). The calculation of formation, degradation and net turnover ratio has been moved from the Results to the Material and Methods section (see lines 239-237 and 315-322).

“Line 322: The procedure of preparing the 3D surface plots should be described in the Method section.”

The description of the preparation of the 3D surface plots has been moved to the Materials and Methods section (see lines 192-193, 370-371, and 393-395).

“Line 325: Describe in the Method section how the LC3B-positive autophagic vacuoles were automatically quantified by using the Image J 325 software.”

The description of the quantification of LC3B-positive autophagic vacuoles has been moved to the Materials and Methods section (see lines 187-189, and 373-374).

“Figure 2 should show the staining also after 0.5, 2 and 24 h, and concomitant bright light images of the cells, to see if these undergo similar changes as shown by TEM.”

The sizes of the cytoplasmic vacuoles we examined are too small for their reliable identification using a light microscope. LC3B immunofluorescence does not seem to be the most suitable method for detecting autophagosomes/autolysosomes, amphisomes, and multilamellar bodies since all of these structures are LC3B-positive. Therefore, we believe the results obtained with electron microscopy are not reproducible using light microscopy and immunofluorescence. However, on TEM images, we have now quantified the autophagosomes/autolysosomes, amphisomes, and multilamellar bodies and hope that these new data provide sufficient evidence to prove the observed shifts in the densities of these cytoplasmic structures (see Figure 4).

“How does PSMα3 affect the metabolic activity and viability of the cells?”

PSMs were shown to exert a cytolytic effect. They can lyse erythrocytes, leukocytes, and other cell types by disrupting the plasma membrane in micromolar concentrations. PSMα contributes to necroptotic cell demise through RIP1/RIP3/MLKL signaling in the THP-1 cell line and human primary macrophages (Kitur K, Parker D, Nieto P, Ahn DS, Cohen TS, Chung S, Wachtel S, Bueno S, Prince A. Toxin-induced necroptosis is a major mechanism of Staphylococcus aureus lung damage. PLoS Pathog. 2015 Apr 16;11(4):e1004820. doi: 10.1371/journal.ppat.1004820. PMID: 25880560.). It has also been demonstrated that PSMα1, 2, and 3 trigger necroptosis by stimulating MLKL phosphorylation in murine neutrophil granulocytes (Zhou Y, Niu C, Ma B, Xue X, Li Z, Chen Z, Li F, Zhou S, Luo X, Hou Z. Inhibiting PSMα-induced neutrophil necroptosis protects mice with MRSA pneumonia by blocking the agr system. Cell Death Dis. 2018 Mar 2;9(3):362. doi: 10.1038/s41419-018-0398-z. PMID: 29500427.). Necroptosis induced by PSMs results in cell lysis, the mechanism of which differs from that resulting from the disintegration of the plasma membrane. Another study demonstrated that PSMα1, and 3 affect cell cycle progression, causing G2/M phase slow-down in the HeLa cell line, this effect, however, was not related to the cytotoxic effect of these toxins Deplanche M, Filho RA, Alekseeva L, Ladier E, Jardin J, Henry G, Azevedo V, Miyoshi A, Beraud L, Laurent F, Lina G, Vandenesch F, Steghens JP, Le Loir Y, Otto M, Götz F, Berkova N. Phenol-soluble modulin α induces G2/M phase transition delay in eukaryotic HeLa cells. FASEB J. 2015 May;29(5):1950-9. doi: 10.1096/fj.14-260513. PMID: 25648996). To our knowledge, the effect of PSMα on the metabolic activity of cells has not yet been investigated in detail. Studies in this direction are extremely promising, as such new data could contribute to a better understanding of the cellular effects of PSMs and the pathogenesis of infections caused by staphylococci.

“Figure 3: Is the large vacuolization indeed due to autophagy? What happens if you add an autophagy inhibitor? The transformation of vacuoles to amphisomes and MLB, suggests that some steps of the autophagy process is interrupt. If it is only stimulation of autophagy, we should see more degradations, but here we have an accumulation of membranes. Can this process be quantified. The images are from one cell. More images are required. It is recommended to perform a time-lapse confocal microscopy using Nile red or another membrane stain to show the entire process and to follow the intracellular fate of the autolysosomes. This would really provide an information about what is going on, and how fast the large autolysosomes are formed. The size of the bars should be shown, and bars of the highest magnification should also be shown.”

We have now quantified the autophagic vacuoles. These new data demonstrate that the proportions of the autophagosomes/autolysosomes, amphisomes, and multilamellar bodies varied significantly within the cytoplasmic vacuole pools over time. Accordingly, we have included a new graph (see Figure 4), modified the Results (see lines 408-411, 414-416 and 418-420), Materials and Methods (see lines 275-280), and the Discussion section (lines 610-614). Furthermore, we also measured the average diameter of autolysosomes, which corresponded to 0.7576±-0.5552 µm (data not shown). We agree with the Referee that some cells indeed contained large autolysosomes. According to previous observations, large autolysosomes can accumulate in the cells during autophagic cell death (Napoletano F, Baron O, Vandenabeele P, Mollereau B, Fanto M. Intersections between Regulated Cell Death and Autophagy. Trends Cell Biol. 2019 Apr;29(4):323-338. doi: 10.1016/j.tcb.2018.12.007. PMID: 30665736.). Since not all cells showed this morphological feature of autophagic cell death, we replaced the 6-h time point image in Figure 4 with one in which the sizes of the autolysosomes better reflect the average measured during our experiments. The sizes of the bars are now shown in Figure 4. Our flux experiments have demonstrated that the autophagic pathway is intact and activated in PSMα3-exposed cells. Previous studies have shown that the autophagic and endolysosomal systems interact closely, and autophagy plays a pivotal role in the formation of multivesicular and multilamellar bodies. In light of previously published observations, we believe that our data correspond to a picture seen during an intensified cooperation between the autophagic and endolysosomal systems. The identification of mechanisms other than autophagy involved in the removal of damaged plasma membrane parts deserves further investigation. However, our present experiments primarily focused on autophagy and used methods accepted in the literature.

 “What is the normal size of autolysosomes? How can you know that these are autolysomes and not autophagosomes?”

Autophagosomes are double-membraned structures of 300-2000 nm in diameter that contain cytoplasmic components. Autolysosomes are single membraned structures of 100-1000 nm that contain degraded cytoplasmic components. Amphisomes are single-membraned structures of 500-1200 nm in diameter that contain undegraded cytoplasmic components and intraluminal vesicles. Multivesicular bodies single-membraned structures of 250-1000 nm in diameter that contain multiple small (50-80 nm) vesicles. Multilamellar bodies are single-membraned structures of 500-1200 nm in diameter that contain concentric membrane layers. Autophagic vacuoles were classified based on these criteria.

“Line 356: AVO should be defined first time used.”

AVO is defined first time used (see line 207).

“Interestingly is the reduced phosphorylation of c-Jun that is also involved in regulating autophagy. Please comment on it.”

We agree with the Referee, and we have now included a comment on c-Jun in the Discussion section (see lines 580-589) and two new references (65 and 66).

“Line 417: Remove hyphen.”

Our language editor recommended the usage of hyphen, thus we would like to keep it.

“Line 478: Please mention in the manuscript, at which concentrations of PSMα3 are lytic.”

The lytic concentration of PSMα3 has now been given (see lines 160-161).

We are grateful for the Referee for the reviewing work and consider that the modifications really do improve the manuscript.

Reviewer 4 Report

Comments and Suggestions for Authors

Dear authors,

your manuscript entitled "Phenol-Soluble Modulin α3 Stimulates Autophagy" addresses a very interesting topic related to the virulence of staphylococci. The role of these kind of modulins has not yet been investigated in relation to autophagy and this is a strong point of this work. The use of different methodologies as western blot analysis, indirect immunofluorescence assay, transmission electron microscopy (TEM) is another strenght of the research. The results are very interesting in keratinocytes and figures, graphs and all the iconographic part are good in quality. The discussion is well structured.

Author Response

Dear Referee 4,

Thank you for your kind message regarding our manuscript entitled "Phenol-Soluble Modulin α3 Stimulates Autophagy." We are delighted to hear that you found our research interesting. Your positive feedback on our use of various methodologies, the quality of our figures and graphs, and the structure of our discussion is greatly appreciated. We express our sincere gratitude for your time and efforts. Your feedback is invaluable to us, and we are encouraged by your positive assessment of our research.

Sincerely yours,

Klára Megyeri M.D., Ph.D.

corresponding author

Round 2

Reviewer 1 Report

Comments and Suggestions for Authors

Dear authors,

thank you for answering the questions and clarifying some issues in the paper. From my side, I can release the paper for publication.

Reviewer 3 Report

Comments and Suggestions for Authors

The authors have done the corrections required.